# Impact of Lightning Data Assimilation on Forecasts of a Leeward Slope Precipitation Event in the Western Margin of the Junggar Basin

**Peng Liu** [1], **Yi Yang** [1,*], **Yu Xin** [2] **and Chenghai Wang** [1]

1   Key Laboratory of Climate Resource Development and Disaster Prevention in Gansu Province, College of Atmospheric Sciences, Lanzhou University, Lanzhou 730000, China; liup16@lzu.edu.cn (P.L.); wch@lzu.edu.cn (C.W.)
2   Institute of Desert Meteorology, China Meteorological Administration, Urumqi 830002, China; learnerxy@163.com
*   Correspondence: yangyi@lzu.edu.cn

**Abstract:** A moderate precipitation event occurring in northern Xinjiang, a region with a continental climate with little rainfall, and in leeward slope areas influenced by topography is important but rarely studied. In this study, the performance of lightning data assimilation is evaluated in the short-term forecasting of a moderate precipitation event along the western margin of the Junggar Basin and eastern Jayer Mountain. Pseudo-water vapor observations driven by lightning data are assimilated in both single and cycling analysis experiments of the Weather Research and Forecast (WRF) three-dimensional variational (3DVAR) system. Lightning data assimilation yields a larger increment in the relative humidity in the analysis field at the observed lightning locations, and the largest increment is obtained in the cycling analysis experiment. Due to the increase in water vapor content in the analysis field, more suitable thermal and dynamic conditions for moderate precipitation are obtained on the leeward slope, and the ice-phase and raindrop particle contents increase in the forecast field. Lightning data assimilation significantly improves the short-term leeward slope moderate precipitation prediction along the western margin of the Junggar Basin and provides the best forecast skill in cycling analysis experiments.

**Keywords:** lightning data assimilation; pseudo-water vapor; leeward slope; precipitation forecast

## 1. Introduction

The Xinjiang Uygur Autonomous Region (hereinafter referred to as Xinjiang) is located in the middle of Asia, is not affected by monsoons and exhibits typical continental climate characteristics [1,2]. In more than 50% of the Xinjiang region, the annual precipitation is below 100 mm [3,4]. Located in the northern part of Xinjiang, the oil industry city of Karamay is situated along the western margin of the Junggar Basin, with the tall Jayer Mountain to the west. Figure 1 shows part of the topography of the Xinjiang region, as well as the height of the terrain near Karamay and the location of the observation sites. Jayer Mountain is a southwest–northeast trending mountain, with a low terrain in the north and a high terrain in the south, and the altitude of the northern part of this mountainous area is approximately 1500 m, while that of the southern part reaches as high as 2400 m. The average annual precipitation in the Karamay region over the past 50 years reaches only 110 mm [5]. A moderate precipitation event with an hourly accumulative precipitation of more than 10 mm lasting for 3 h occurred on the western margin of the Junggar Basin on 4 July 2013. Figure 2 shows the observed 1-h accumulated precipitation levels from 1200 to 1300 UTC, 1300 to 1400 UTC and 1400 to 1500 UTC on 4 July 2013. This precipitation event would be difficult to consider a heavy precipitation event in a monsoon area with abundant rainfall. However, it certainly caused severe economic losses in arid areas. Timely and

accurate forecasting and warning of disastrous precipitation weather conditions in arid regions are critical to safeguard properties.

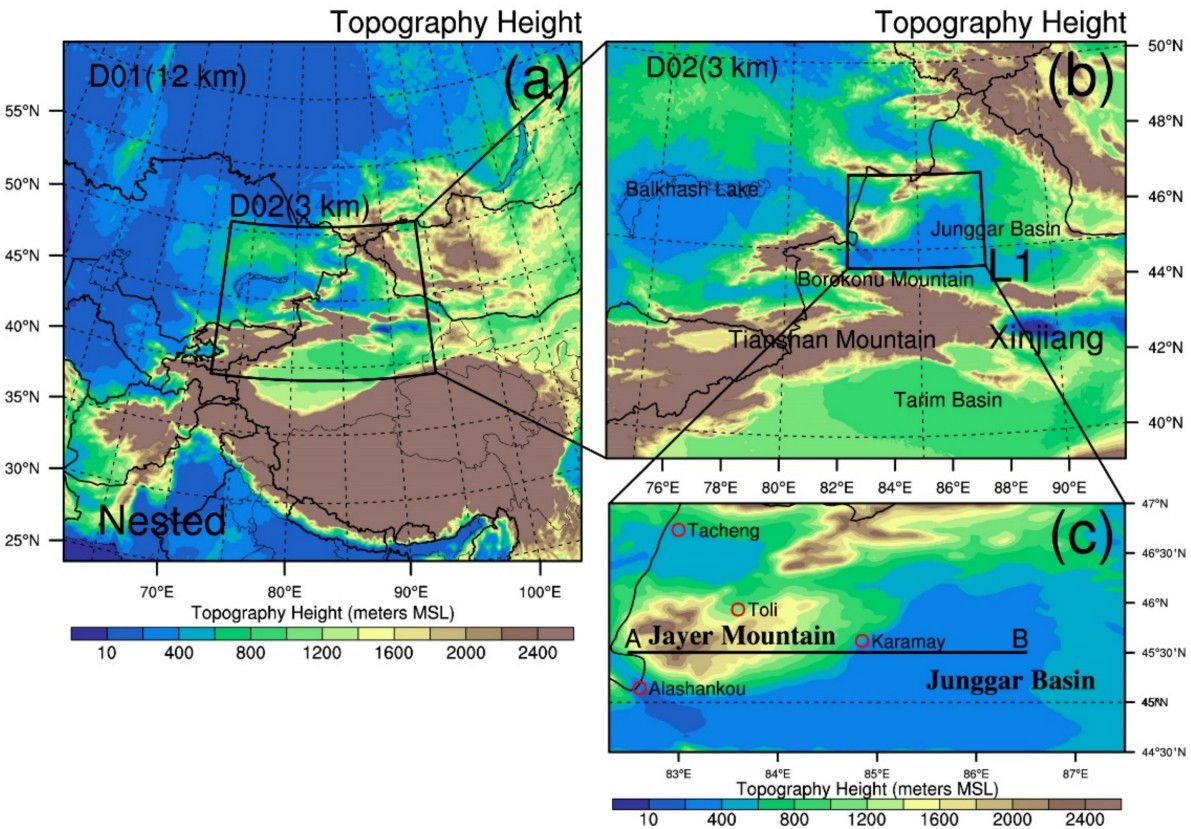

**Figure 1.** Model terrain height in the WRF model domain. (**a**) Nested D01 and D02 model domain. D01 and D02 with grid spacings of 12 and 3 km, respectively. (**b**) D02 model domain with a grid spacing of 3 km. The solid black rectangular box (L1) in (**b**) indicates the location of focus. (**c**) The location of the national ordinary station (red hollow circles). The black line AB in (**c**) indicates the location of the vertical cross-section in the subsequent figures.

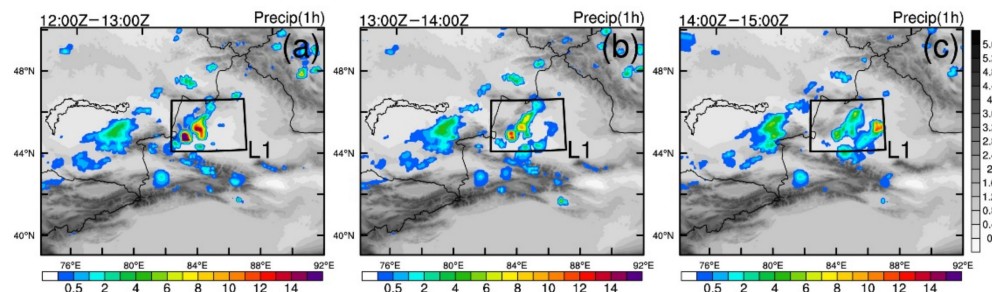

**Figure 2.** Observed 1-h accumulated precipitation levels from 1200 to 1300 UTC (**a**), 1300 to 1400 UTC (**b**) and 1400 to 1500 UTC (**c**) on 4 July 2013. The shades of gray in each plot indicate the model terrain height.

Accurately forecasting heavy precipitation events on leeward slopes is a challenge due to the influence of downdrafts upon crossing the mountains [6,7]. Currently, data assimilation methods that reduce initial field errors comprise the main strategy to improve the numerical weather prediction (NWP) skill [8–10]. Lightning, as a phenomenon that occurs within thunderstorm clouds, can identify areas of deep, mixed-phase convection [11–16]. In thunderstorm clouds, various types of charged hydrated particles collide under the action of air currents and gravity to produce discharge. Relying on extensive ground-based networks and spaceborne optical instruments, lightning data have the advantages of high real-time performance, wide coverage and low terrain influence.

Many studies have verified that lightning data assimilation can greatly improve forecasts of severe convective precipitation. The assimilation of rain rates derived from satellites and lightning suggests that lightning data have a positive impact on super-storm forecasts [17–19]. The study of assimilating lightning data to a mesoscale model by Papadopoulos et al. showed that the assimilation of lightning data can significantly improve the model's prediction accuracy of convective precipitation in the assimilation period [20,21]. Mansell et al. used lightning data to control the Kain–Fritsch (KF) convection parameterization scheme. The results showed that lightning assimilation was successful in generating cold pools that were present in the surface observations at forecast initialization. In the next study, they assimilated the total lightning flash extent data using the ensemble filter method [22,23]. Fierro et al. applied a total lightning data assimilation technique for tornado outbreak forecasting. The assimilation of total lightning data significantly improved the representation of convection at the analysis time and better depicted individual supercell structures at the 1-h forecast [24–26]. Qie et al. proposed a total lightning data assimilation method and applied it in several thunderstorm processes over northern China. The representation of convection was significantly improved 1 h after lightning data assimilation and even during the assimilation period [27]. Gan et al. and Liu et al. applied lightning to several severe convective events in eastern China. The results showed that assimilating lightning data effectively improves the forecasting of convective precipitation [26,28].

Lightning data are generally assimilated into a model based on a lightning proxy variable that drives the relationships between lightning and other model variables. Based on the relationship between the lightning flash density and rainfall, the simulated latent heat profile is adjusted according to lightning observations, which positively impacts the forecast performance [17–19]. However, there is no universal relationship between lightning and rainfall, and the relationship between lightning and rainfall is sensitive to storm location and season [17,29–31]. By establishing an empirical relationship between the lightning frequency and vertical velocity, the vertical velocity proxy variable created from lightning data is assimilated into the model with the dynamic nudging and ensemble square root filter (EnSRF) methods [32,33]. The empirical relationships between vertical velocity or temperature and lightning are limited by local climate characteristics and differences in convective processes. The proxy radar reflectivity driven by lightning data can also be assimilated [34–37]. However, not only are there errors in the proxy radar reflectivity from lightning data, but the radar data assimilation system itself has many shortcomings. Based on the physical mechanism of lightning occurrence, an empirical relationship between ice-phase particles or the graupel content and the flash extent density (FED) can be established, and the FED is then assimilated by nudging or ensemble methods [27,38–42]. This assimilation method allows a direct update of the hydrometeor variables in the analysis field but lacks adjustment for the water vapor environment, and the innovation of ice and graupel is usually maintained for a limited time in the forecast. Researchers have applied lightning data assimilation methods based on the cloud saturation assumption inferred from the electrical state. Pseudo-water vapor observations have been created and assimilated through the 3DVAR method, which has effectively improved the forecasting skill of severe convective precipitation [24–26,43]. Many notable results have been reported with the 3DVAR method [44–49]. The assimilation scheme based on water vapor is a simple and effective method. For different convective events, there may be a wet bias in the forecast field due to the errors in the obtained pseudo-water vapor observations [24,26].

However, current lightning data assimilation studies have mostly been conducted in monsoon-influenced regions where severe convection frequently occurs, considering thermal field adjustment schemes based on latent heat profiles, dynamic field adjustment schemes of the vertical velocity and hydrometeor variables describing the radar reflectivity and graupel and ice-phase particles or water vapor adjustment schemes. Regarding the arid Xinjiang region, which exhibits convective attributes that differ from those of monsoon-

influenced regions, the impact of the assimilated lightning data on convective precipitation forecasts is an issue worth investigating. In addition, Xinjiang contains few radar sites for the detection of strong convection, and the application of lightning data can better complement the lack of observations of strong convective precipitation.

In particular, the impact of lightning data on the prediction of leeward slope precipitation events influenced by topography has been little studied, e.g., whether lightning data assimilation can improve the forecasting capability of precipitation on leeward slopes or whether it can provide a better understanding of the development mechanism and physical process of leeward slope moderate precipitation through numerical simulation. In this study, a moderate precipitation event occurring in Karamay on 4 July 2013 was analyzed. Single and cycling experiments were performed to evaluate the impact of lightning data assimilation.

Based on the above discussion, this work focuses on the impact of lightning data assimilation on the leeward slope of the Xinjiang region, providing a scientific basis for lightning data to compensate for radar observations in this region. Furthermore, based on the numerical forecast results when assimilating lightning data, the occurrence mechanism and spatial structure of this precipitation event were analyzed to provide guidance in the forecasting of future leeward slope precipitation events in this region.

In the second section of the manuscript, the experimental design, data and methods used in this study are introduced. The synoptic description is described in the third section. In the fourth section, the results are presented. The last section provides the conclusions and discussion.

## 2. Data and Methods

### 2.1. Experimental Design

In this study, a moderate precipitation event that occurred on Jayer Mountain in northern Xinjiang was selected to examine the performance of lightning data assimilation. A diagram of the experimental design is shown in Figure 3. A control run (CTL) was initialized at 0600 UTC on 4 July 2013 with a 6-h spin-up period. The control run has no data assimilation and forecast for 3 h to 1500 UTC. A single analysis experiment (LDA) was conducted to gain a preliminary understanding of the effect of lightning data assimilation on moderate precipitation prediction in Jayer Mountain, and a cycling analysis experiment (CLDA) was conducted to verify the performance of lightning data in cyclic assimilation. The single and cycling analysis experiments were also initialized at 0600 UTC. In the single analysis experiment, the forecast results at 1200 UTC were used as the background to perform a single lightning data assimilation. Then, the single analysis field was used to initiate forecasts for 3 h to 1500 UTC. In the cycling analysis experiment, the forecast results at 1100 UTC were used as the background for the first analysis, and the analysis field was used as the initial field for the 1-h forecast. After the first analysis, the forecast results at 1200 UTC were used as the background for the second analysis. Then, the second analysis field was used to initiate forecasts for 3 h to 1500 UTC.

The three-dimensional compressible nonhydrostatic Weather Research and Forecasting (WRF) model version 4.2 (WRF4.2) was adopted in this study. All experiments were conducted over the double-nesting domain. D01 has a 351 × 351 grid with a grid spacing of 12 km (Figure 1a). D02 has a 501 × 401 grid with a grid spacing of 3 km (Figure 1b). In the vertical direction, there were 51 terrain-following eta layers, and the model top was 10 hPa. The analysis is performed only on the D02 domain. The initial and boundary conditions were provided by the National Centers for Environmental Prediction Final Operational Global Analysis data (FNL) at 6-h intervals and a 1° × 1° grid spacing. The physics schemes included the Thompson microphysical parameterization [50], the Dudhia scheme for shortwave radiation [51], the Rapid Radiative Transfer Model scheme (RRTM) for longwave radiation [52], the Yonsei University planetary boundary layer scheme [53] and the Unified Noah land surface model [54]. To better resolve the convective processes,

the Kain–Fritsch (KF) cumulus parameterization scheme [55] was used in the D01 domain. In this work, only the results of D02 are shown.

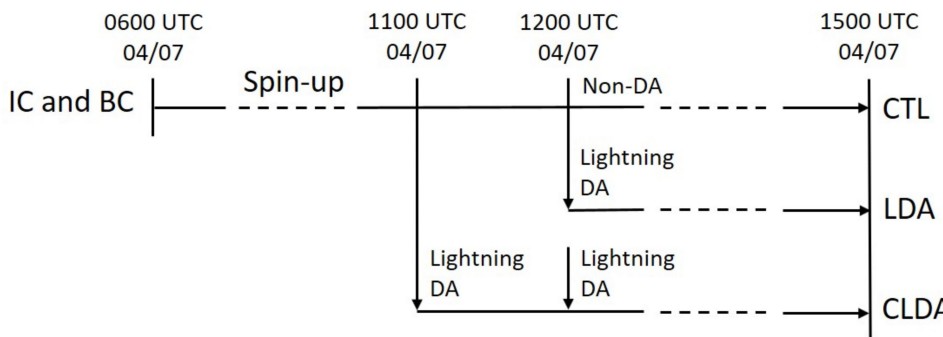

**Figure 3.** A diagram of the experimental design.

### 2.2. Data Used for Assimilation and Validation

In this study, assimilated lightning data were obtained from the Advance Direction and Time-of-Arrival Detecting (ADTD) cloud-to-ground lightning location system in Xinjiang. In the location system, the joint method of the directional time difference was used to locate lightning. The lightning location system can provide elements such as time, latitude, longitude and intensity of the occurrence of cloud-to-ground lightning and cover the whole territory of Xinjiang and surrounding areas. Quality control of the lightning data was performed prior to assimilation to remove potentially duplicate lightning strikes. If the time interval between two lightning records was shorter than 100 ms and the distance between two adjacent points was smaller than 7 km, then the corresponding two records were counted as the same lightning record. The lightning data used in this study ranged from 1030 UTC to 1230 UTC on 4 July 2013.

The $0.1° \times 0.1°$ hourly precipitation product combined with site precipitation observations and the Climate Prediction Center morphing method (CMORPH) precipitation product provided by the Chinese National Meteorological Information Center (NMIC) were employed to evaluate the precipitation forecast skill. This product is based on the global 30-min and 8-km resolution CMORPH satellite inversion precipitation product developed by the U.S. Climate Prediction Center following quality control and correction of hourly precipitation data from 30,000 automated weather station observations in China. An hourly precipitation product with a resolution of $0.1°$ has an overall error level within 10% and an error level for heavy precipitation and regions with sparse stations within 20% [56–58]. The accumulated precipitation at the model grid points was interpolated onto the precipitation product grid points before skill score calculation. The weighting method used was the simple inverse distance squared.

### 2.3. 3DVAR Method

The 3DVAR method, which can assimilate multiple types of data, was implemented in this study. Compared with the ensemble method, the more advanced four-dimensional variational (4DVAR) method or hybrid method, the 3DVAR method has a low computational cost. The cost function $J(\mathbf{x})$ of the 3DVAR method is as follows [59]:

$$J(\mathbf{x}) = J_b(\mathbf{x}) + J_o(\mathbf{x}) = \frac{1}{2}\left(\mathbf{x} - \mathbf{x}^b\right)^T \mathbf{B}^{-1}\left(\mathbf{x} - \mathbf{x}^b\right) + \frac{1}{2}(\mathbf{y} - \mathbf{y}^o)^T \mathbf{O}^{-1}(\mathbf{y} - \mathbf{y}^o), \qquad (1)$$

where $\mathbf{x}$ is the analysis state, $\mathbf{x}^b$ and $\mathbf{y}^o$ are the background and observation, respectively, and $\mathbf{B}$ and $\mathbf{O}$ are the background and observation error covariance matrix, respectively. In general, Equation (1) is converted into an incremental form of the control variables. In this study, the control variables are the stream function, unbalanced velocity potential, unbalanced temperature, pseudo-relative humidity and unbalanced surface pressure. The background error covariance $\mathbf{B}$ is estimated with the National Meteorological Center (NMC)

method [60]. The statistics are estimated with the differences of 24 and 48-h GFS forecasts with T170 resolution, valid at the same time for 357 cases, distributed over a period of one year. In the assimilation procedure, the determination of the observation error is very important for the analysis results. As this study considered a new observation type, namely, pseudo-water vapor, it was not possible to obtain accurate statistics on the observation error, so the bogus type of observation error in the WRF data assimilation system (WRFDA) was referenced. The observation error of relative humidity for bogus types from 850 to 100 hPa is 10% in WRFDA. In order to alleviate possible bad impact of pseudo-observation on the analysis field, the observation error of the relative humidity from 850 to 100 hPa is 15% in this study, which is an increase of 5% compared to that in WRFDA default. This setting is the same as in Liu et al. [26].

### 2.4. Lightning-Driven Pseudo-Water Vapor

As mentioned in the introduction, the assimilation scheme based on water vapor was implemented in this study, using a similar procedure to that of Liu et al. [26]. Pseudo-water vapor observations were obtained by adjusting the model background relative humidity field using the lightning frequency. The lightning data were first unified to the nearest model grid point, that is, the lightning was categorized to the nearest pattern grid point to it. Then, the unified lightning data were accumulated in 1-h intervals between 30 min before and 30 min after the analysis time. Figure 4a,b show the accumulated lightning frequency from 1030 to 1130 UTC and from 1130 to 1230 UTC on 4 July 2013, respectively. At each model grid point where the accumulated lightning frequency exceeded zero, the relative humidity determined from the background was adjusted to 90% within an assumed fixed depth of 3 km above the lifted condensation level (LCL, a surrogate for cloud base). In particular, when the relative humidity was higher than or equal to 90% at that location, no adjustment was performed. The relative humidity values obtained after lightning data adjustment were assimilated as pseudo-water vapor observations with the 3DVAR method.

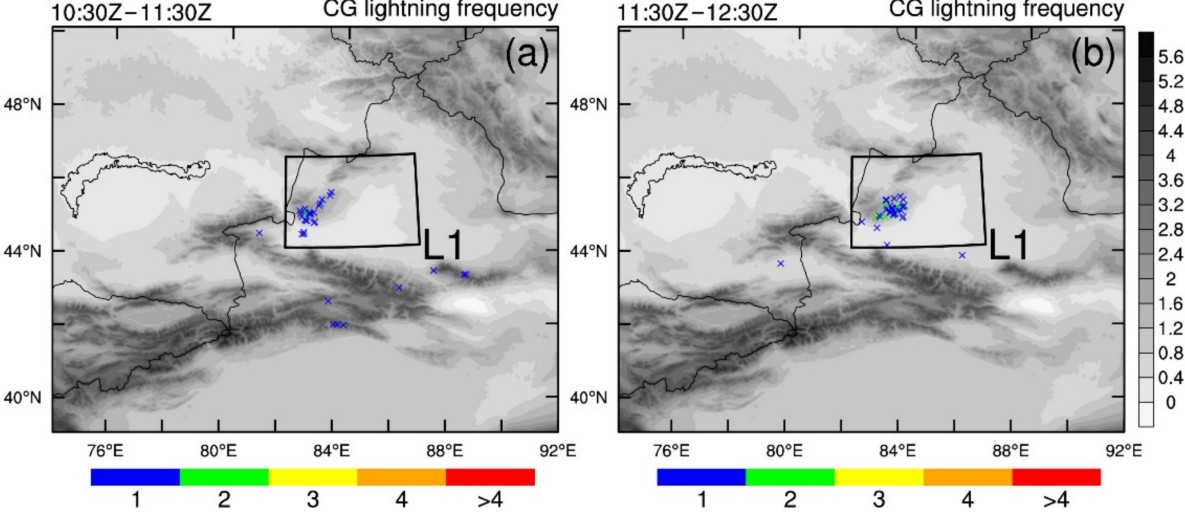

**Figure 4.** The accumulated lightning frequency from 1030 to 1130 UTC (**a**) and from 1130 to 1230 UTC (**b**) on 4 July 2013. The shades of gray in each plot indicate the model terrain height.

## 3. Synoptic Description

At 1200 UTC on 4 July 2013, a moderate precipitation event accompanied by lightning occurred along the western margin of the Junggar Basin. This precipitation event had an hourly accumulative precipitation of more than 10 mm and lasted for three hours (Figure 2). Although it was a light-to-moderate rainfall event, it caused very serious damage to the local economy and property in arid areas. The rainfall center is indicated by a black box in Figure 1b (labeled L1). The 1200 UTC analysis results on 4 July 2013 from ERA5 hourly data

are shown in Figure 5. This moderate precipitation event occurred against the background of a deep cold low vortex in the northern part of Balkhash Lake. At 1200 UTC, the surface temperature was higher in the western part of L1, with a small depression in the dew point (Figure 5a). At 850 hPa, there was a strong westerly and southwesterly flow near Balkhash Lake and a northwesterly over-mountain flow in the L1 region under the influence of Jayer Mountain (Figure 5b). At 700 hPa, southern L1 provided sufficient water vapor conditions, and weak shear occurred in the wind field influenced by the nearby mountains (Figure 5c). At 500 hPa, there was a shallow trough line in the western part of L1 (Figure 5d).

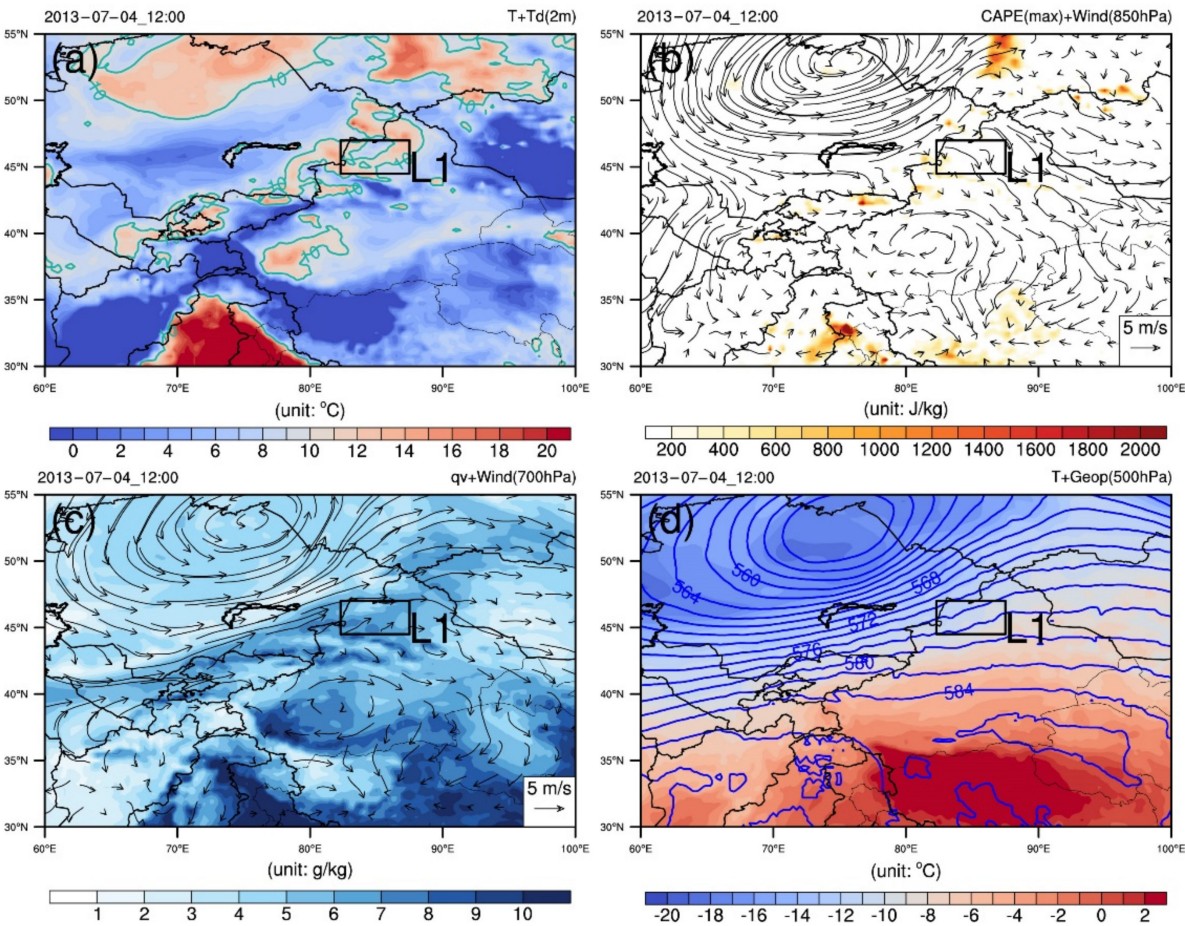

**Figure 5.** ERA5 hourly reanalysis data at 1200 UTC on 4 July 2013. (**a**) The 2-m temperature (T, colored contours, °C) and 2-m dewpoint temperature (Td, light-sea-green contour lines at 10 °C). (**b**) Maximum convective available potential energy (CAPE, colored contours, J/kg) and 10-m horizontal wind (wind vector). (**c**) Specific humidity (qv, colored contours, g/kg) and horizontal wind (wind vector) at 750 hPa. (**d**) Temperature (T, colored contours, °C) and geopotential height (Geop, blue line contours, gpm) at 500 hPa.

To provide a better understanding of the background environment of this moderate precipitation event, Figure 6 shows the skewed T-logp sounding results obtained at the Tacheng and Karamay sites. The geographical locations of the Tacheng and Karamay sites are shown in Figure 1c. The Tacheng site is located on northern Jayer Mountain, and the altitude is 535 m. At 1200 UTC, the Tacheng site was in the eastern part of the deep cold low vortex so that the wind was southwesterly in the middle and high levels at the station. At the Tacheng site, a lower lift condensation height and higher convective available potential energy (CAPE) occurred (Figure 6a). The Karamay site is located in the eastern foothills of Jayer Mountain, and the altitude is 445 m. At the Karamay site, above 700 to 500 hPa, the dew point temperature is closer to the temperature profile (Figure 6b).

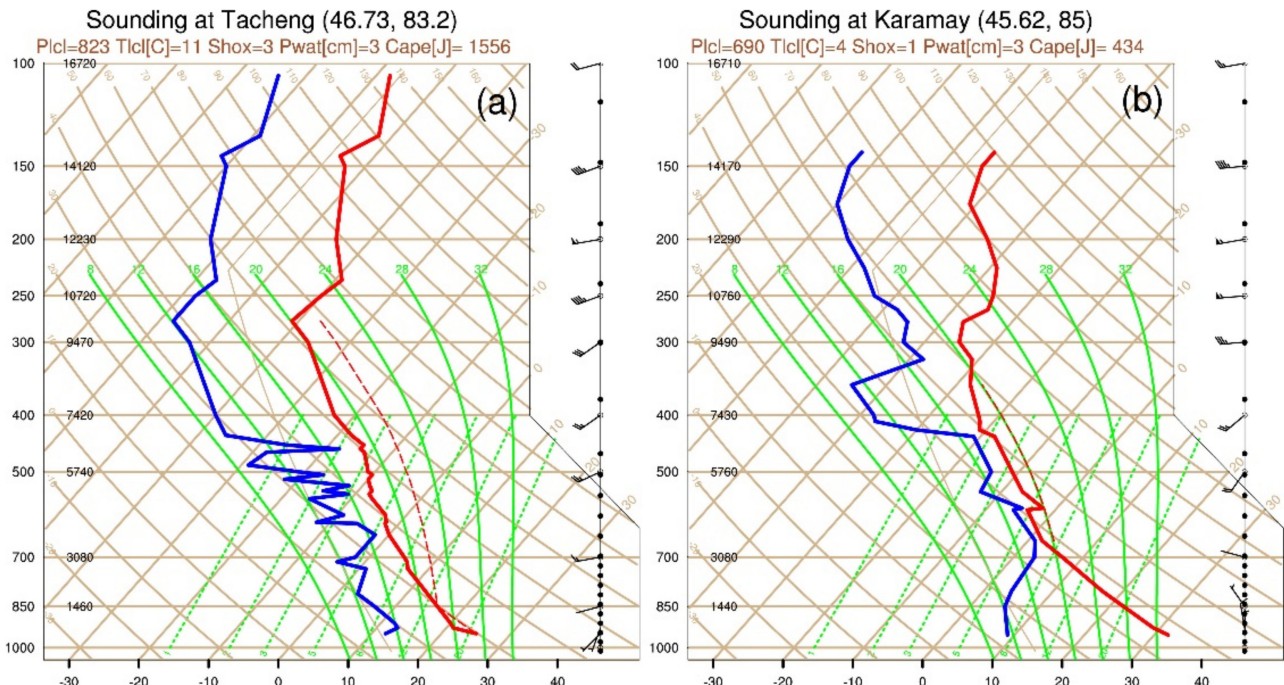

**Figure 6.** Skewed T-logp plots at 1200 UTC on 4 July 2013 at the (**a**) Tacheng sounding site and (**b**) Karamay sounding site.

## 4. Results

In this section, the impact of lightning data assimilation on the analysis field is presented. Since the analysis field changes, the dynamic and thermal variables in the forecast field are analyzed. Finally, the performance of short-term precipitation forecasting is evaluated based on subjective precipitation patterns and locations and objective skill scores.

### 4.1. Analysis Fields

Adequate water vapor conditions are necessary for moderate precipitation to occur. Figure 7 shows horizontal and vertical cross-sections of the relative humidity in the control run and the analysis field of the lightning assimilation experiments. In the control run, the relative humidity was low at L1, exceeding 70% at only a few locations. In both the single and cycling analysis experiments at 1200 UTC, the determined analysis field revealed a relative humidity higher than 80% in the Jayer Mountain region. Since the cycling analysis experiment assimilated more lightning data, the cycling experiment obtained a higher relative humidity over a larger range than did the single analysis experiment. As shown in the vertical cross-section of the relative humidity (Figure 7d–f), the cycling analysis experiment determined a higher relative humidity in the near ground layer on the leeward slope.

In the analysis, the relative humidity was effectively adjusted via the assimilation of pseudo-water vapor-driven lightning data, and large relative humidity increments were obtained on the leeward slope. Compared with the control experiment, the assimilation experiments more favorably triggered convection under near-saturated water vapor conditions on the leeward slope.

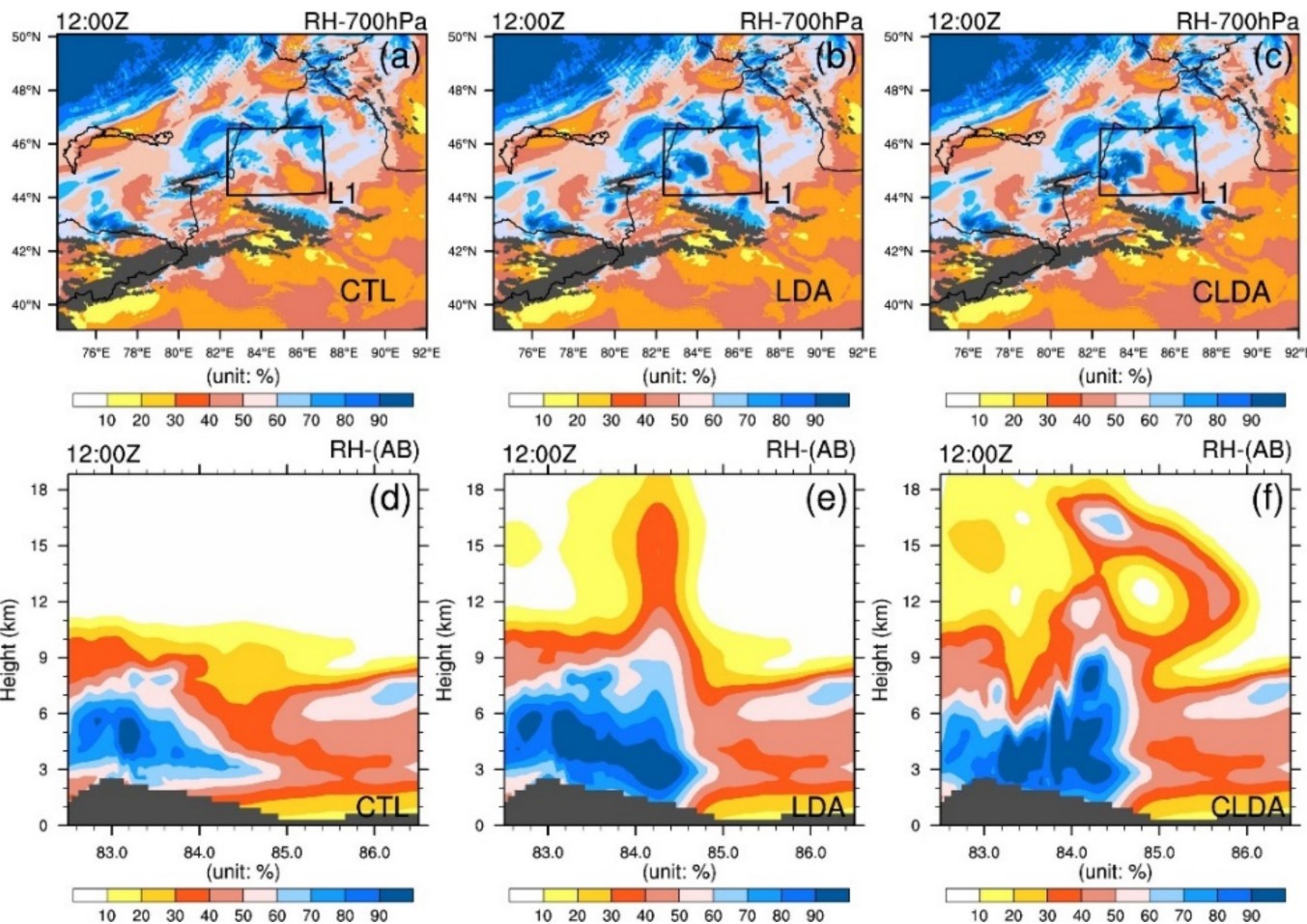

**Figure 7.** Relative humidity in the control run and analysis results of the assimilation experiments at 1200 UTC on 4 July 2013. (**a–c**) Horizontal cross-section at 700 hPa. (**d–f**) Vertical cross-section along line AB in Figure 1c. (**a,d**) Control run (CTL), (**b,e**) single analysis experiment (LDA) and (**c,f**) cycling analysis experiment (CLDA).

### 4.2. Forecast Field

As the water vapor content in the analysis field changes, this should lead to a corresponding change in the forecast. After 1 h of forecasting, the over-mountain flow from Jayer Mountain in the assimilation experiments was strengthened. The transmountain flow from the northwest and southeast caused a weak convergent motion in the L1 region with the northerly flow on the northeast side of L1 (Figure 8b,c). With strengthening transmountain flow, wet and cold air was transported from the windward slope to the leeward slope, resulting in a temperature decrease near Jayer Mountain in the assimilation experiments (Figure 8b,c). At 700 hPa, there was a significant increase in the water vapor mixing ratio in the southwestern part of L1 in the assimilation experiments (Figure 8e,f). At 500 hPa, both the control and assimilation experiments were dominated by southwesterly winds, and the wind speed increased at L1 in the assimilation experiments (Figure 8g–i). In the assimilation experiments, positive and negative concomitant potential vorticities were found in the L1 region. The anomalous variation in the potential vorticity at this position was associated with the vertical motion.

Figure 9 shows the vertical thermodynamic conditions in the forecast field (1-h) along line AB in Figure 1c. In the assimilation experiments, obvious upward and downward movement was found at the bottom of the leeward slope with a higher relative humidity in the middle layer (Figure 9b,c). In addition, the assimilation experiments indicated positive and negative adjacent potential vorticity changes on the leeward slope, and the high-value zone of the pseudo-equivalent potential temperature extended from the upper to the

middle and lower layers at the location of significant vorticity changes (Figure 9e,f). In the single analysis experiment, the upward movement corresponds well to positive potential vorticity, and the downward movement corresponds well to negative potential vorticity (Figure 9d,e). A high-water vapor environment is a necessary condition on leeward slopes for precipitation to occur, as the unstable energy in the bottom layer is transported toward the middle layers under vertical motion, and wet air uplift on the leeward slope greatly facilitates precipitation occurrence.

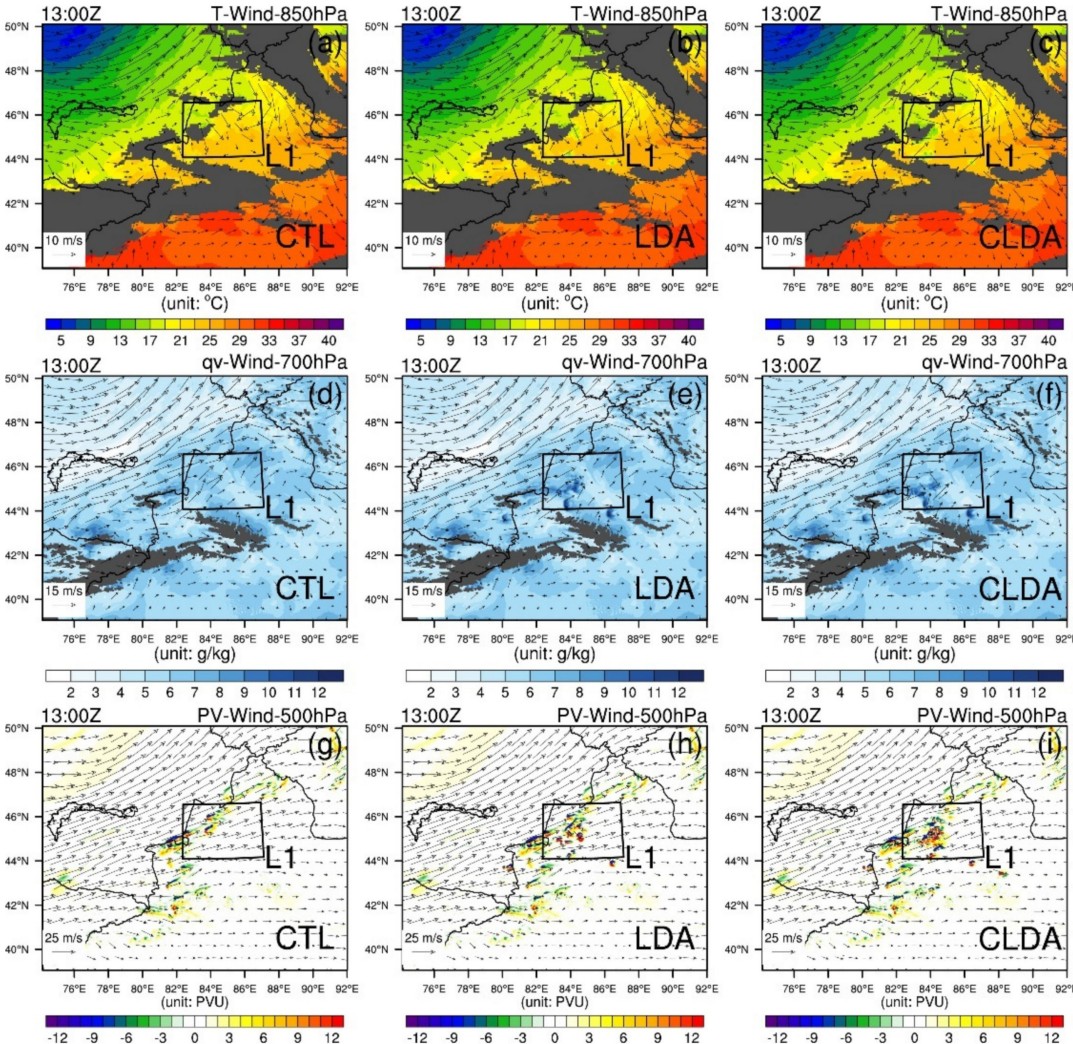

**Figure 8.** Horizontal cross-section of the temperature and wind vector at 850 hPa (**a**–**c**), water vapor mixing ratio and wind vector at 700 hPa (**d**–**f**), potential vorticity and wind vector at 500 hPa (**g**–**i**) at (1-h forecast) 1300 UTC on 4 July 2013. (**a**,**d**,**g**) Control run (CTL), (**b**,**e**,**h**) single analysis experiment (LDA) and (**c**,**f**,**i**) cycling analysis experiment (CLDA). The shades of gray in (**a**–**f**) indicate the missing values due to the terrain.

Figure 10 shows the temperature and content of hydrometeor variables at the different levels, including the graupel mixing ratio ($q_g$), snow mixing ratio ($q_s$) and rain mixing ratio ($q_r$). The water vapor content increased west of L1 after lightning data assimilation, and rain droplets and ice crystal particles increased after a 1-h forecast period under the combined effect of the humidity, temperature and dynamic field. In the central part of L1, $q_g$ and $q_s$ exceeded 1 g/kg in the assimilation experiments at 500 hPa, while the control run only revealed low $q_s$ values in the northwestern part of L1. At 700 hPa, $q_r$, which is the most relevant for precipitation, exceeded 1 g/kg in the west-central part of L1 in the assimilation experiments, while $q_r$ was lower than 0.3 g/kg across the entire L1 region in the control run. Similar variations in the hydrometeor variables are shown more clearly in the vertical cross-section (Figure 11). There was a significant increase in $q_g$ and $q_s$ in the upper layers and an increase in $q_r$ in the lower layers in the assimilation experiments. The increase in the hydrometeor variables was greater in the cycling analysis experiment than in the single analysis experiment. Since latent heat is released upon water condensation, the temperature accordingly increased at the locations where the hydrometeor variables increased.

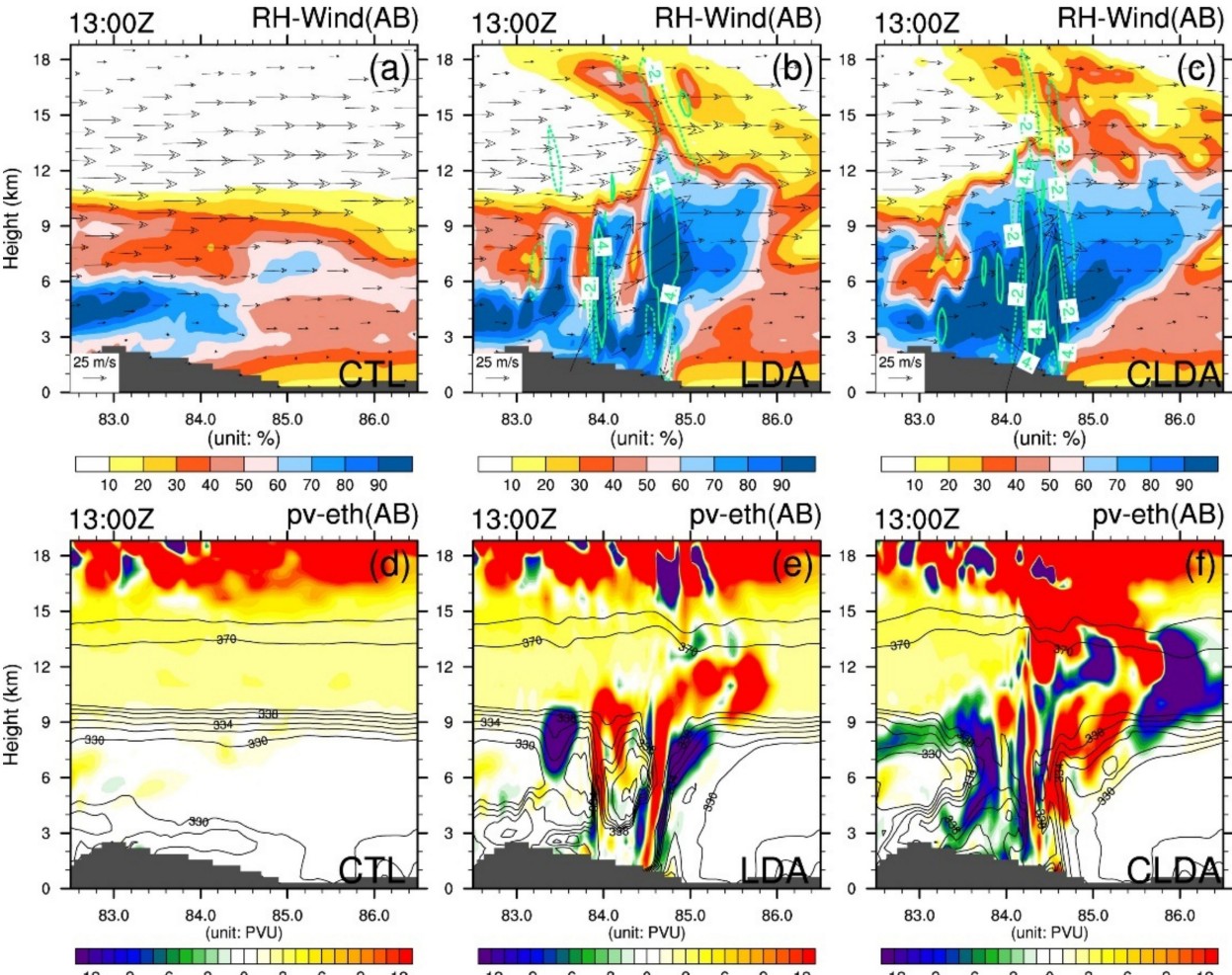

**Figure 9.** Vertical cross-section of the relative humidity and wind vector (**a–c**), potential vorticity and equivalent potential temperature (**d–f**) at (1-h forecast) 1300 UTC on 4 July 2013 along line AB in Figure 1c. (**a,d**) Control run (CTL), (**b,e**) single analysis experiment (LDA) and (**c,f**) cycling analysis experiment (CLDA). The dashed/solid green line in (**b,c**) indicates a vertical velocity of −2/4 m/s. The shades of gray in each plot indicate the model terrain.

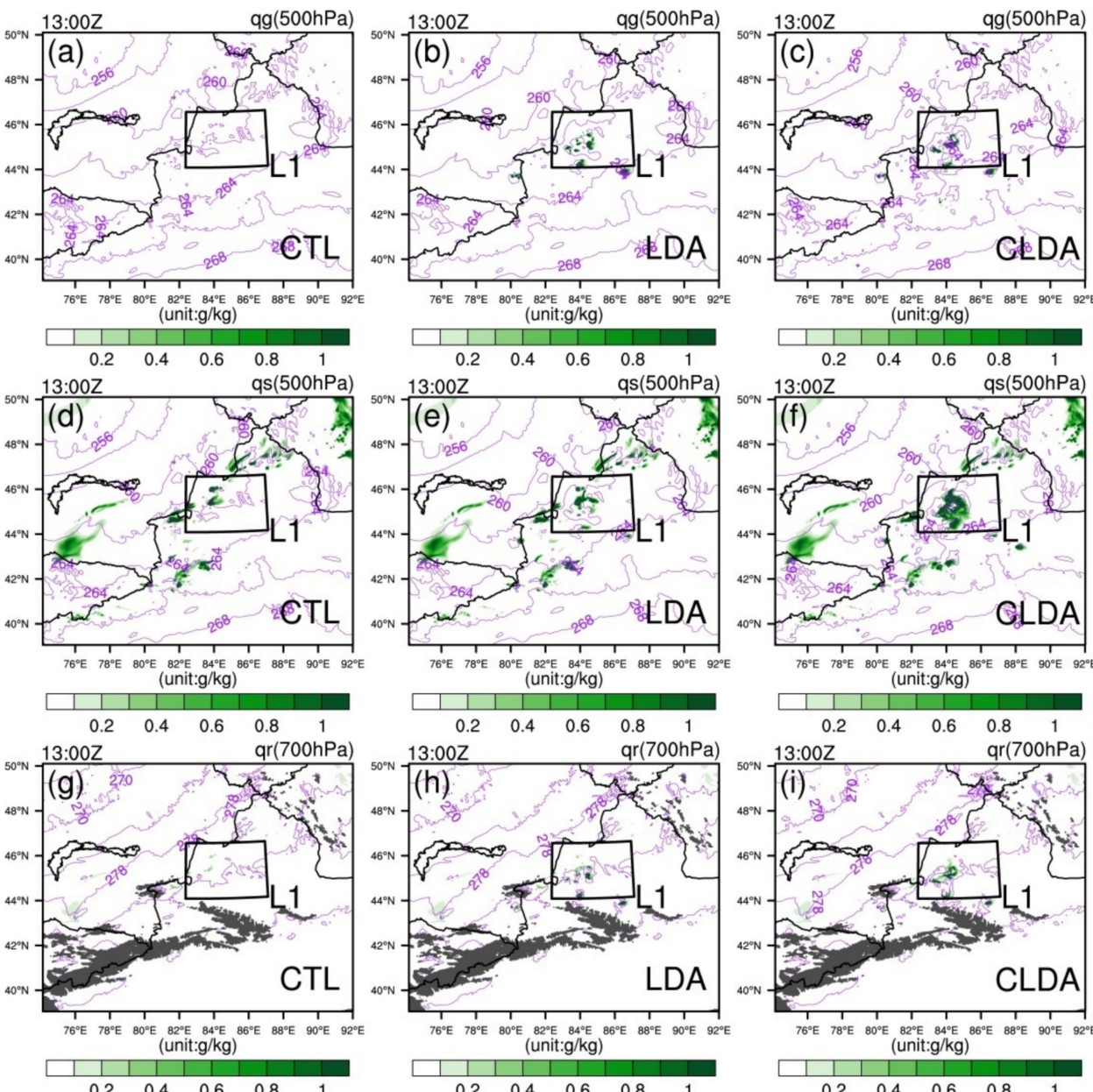

**Figure 10.** Temperature (the purple contour lines) and graupel mixing ratio ($q_g$) at 500 hPa (**a–c**), snow mixing ratio ($q_s$) at 500 hPa (**d–f**) and rain mixing ratio ($q_r$) (**g–i**) at 700 hPa (1-h forecast) at 1300 UTC on 4 July 2013. (**a,d,g**). Control run (CTL), (**b,e,h**) single analysis experiment (LDA) and (**c,f,i**) cycling analysis experiment (CLDA). The shades of gray in (**g,h,i**) indicate the missing values due to the terrain.

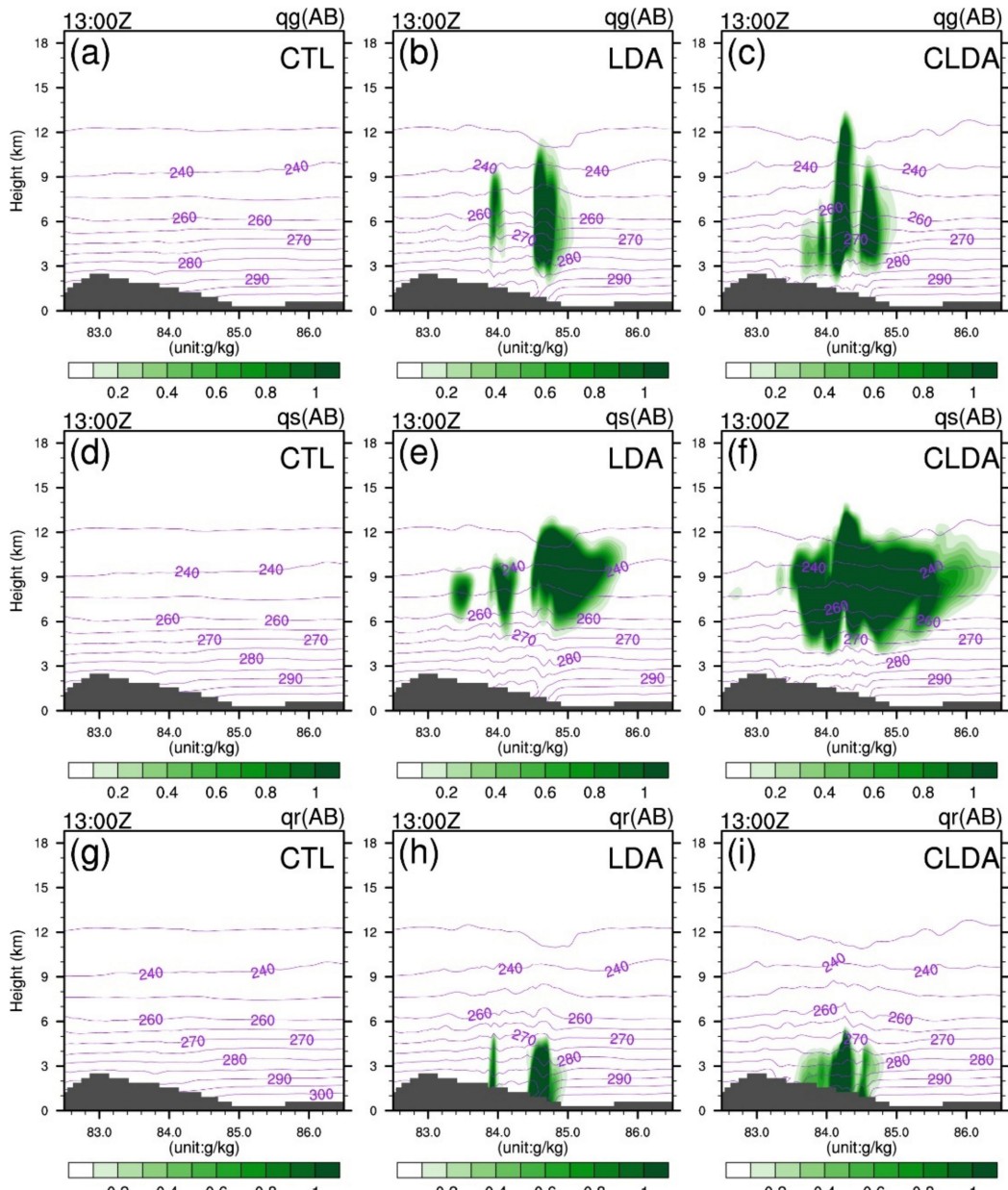

**Figure 11.** Vertical cross-section of the temperature (the purple contour lines), graupel mixing ratio ($q_g$) (**a–c**), snow mixing ratio ($q_s$) (**d–f**) and rain mixing ratio (**g–i**) at the 1-h forecast at 1300 UTC on 4 July 2013 along line AB in Figure 1c. (**a,d,g**) Control run (CTL), (**b,e,h**) single analysis experiment (LDA) and (**c,f,i**) cycling analysis experiment (CLDA). The shades of gray in each plot indicate the model terrain.

### 4.3. Precipitation Evaluation

The precipitation forecast performance after lightning data assimilation along the western margin of the Junggar Basin was evaluated based on subjective spatial distribution patterns and objective skill scores. An equitable threat score (ETS) equal to 1 suggests a perfect forecast, and an ETS equal to 0 suggests the absence of forecast skill. A performance diagram conveniently combines information on the frequency bias, probability of detection (POD), critical success index (CSI) and success ratio (1 minus the false alarm rate) [61].

The observed and forecasted 3-h accumulated precipitation levels are shown in Figure 12. From 1200 UTC to 1500 UTC, a strong precipitation band occurred in the northwestern part of L1 (Figure 12a). The control run could not forecast the precipitation occurring at this location. The assimilation experiments could better forecast the precipita-

tion band, and the precipitation locations were almost identical to the observation locations, while the precipitation patterns were similar. The cycling analysis experiment predicted a greater precipitation magnitude and range than did the single analysis experiment because the relative humidity increment was the largest. A weak rainfall band occurred in the southeastern part of L1. No lightning occurrence was observed at this location, and the assimilation experiments did not provide a better precipitation forecast at this location.

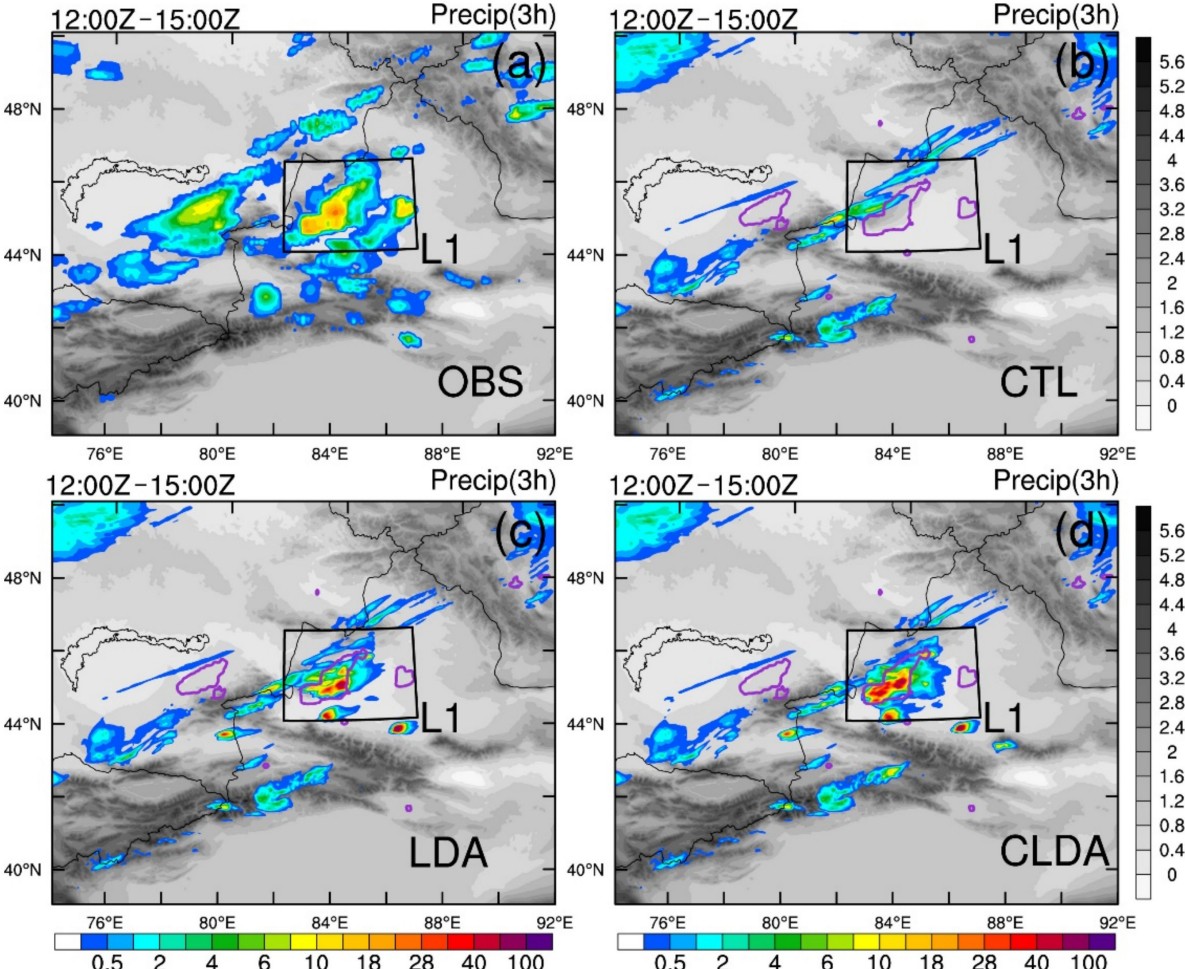

**Figure 12.** Observed and forecasted 3-h accumulated precipitation levels from 1200 to 1500 UTC on 4 July 2013. The shades of gray in each plot indicate the model terrain height. (**a**) Observations (OBS), (**b**) control run (CTL), (**c**) single analysis experiment (LDA) and (**d**) cycling analysis experiment (CLDA). The contours of the observations (5 mm) are highlighted with a dark orchid line in (**b**–**d**).

Figure 13 shows the ETS of the forecasted hourly accumulated precipitation in the L1 region. Consistent with the results of the spatial distribution of the accumulated precipitation, the control run achieved no forecast skill at 1 and 5 mm. In the assimilation experiments, the ETS approached 0.2 at lower thresholds (0.1 and 1 mm) and was higher than 0.2 during 1200 to 1400 UTC at the 5-mm threshold level. In particular, the ETS of the cycling analysis experiment was higher than that of the single experiment at almost all times of the respective threshold. In the performance diagram, the cycling analysis experiment also exhibited a higher POD and success ratio (Figure 14). Consistent with the ETS score results, there is still no forecast skill for the control run. Overall, assimilation experiments have a probability of detection near 0.4 and a success ratio near 0.6 at smaller thresholds (0.1 and 1 mm) (Figure 14a,b). At each threshold, the point of the cycling experiment is always to the upper right of the single experiment, which indicates that the

cycling experiment has a higher probability of detection and success ratio than the single analysis experiment.

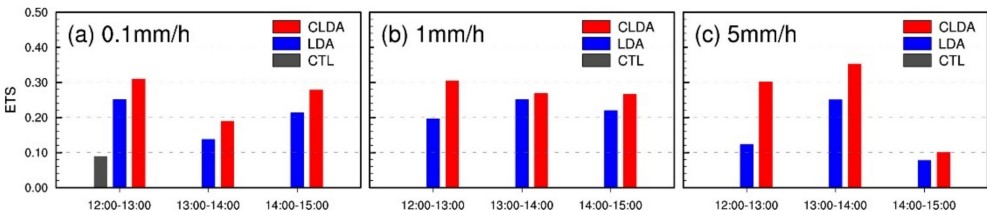

**Figure 13.** Equitable threat score (ETS) of the forecasted hourly accumulated precipitation at L1 from 1200 to 1500 UTC on 4 July 2013. (**a**) The 0.1-mm threshold, (**b**) 1-mm threshold and (**c**) 5-mm threshold.

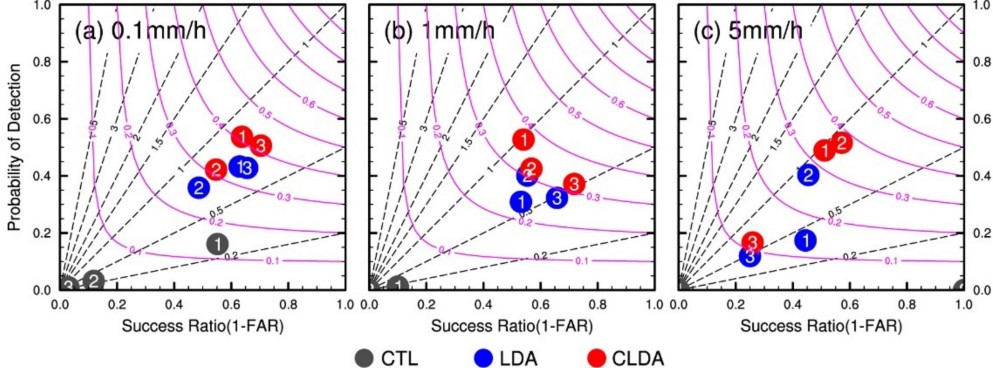

**Figure 14.** Performance diagram of the forecasted hourly accumulated precipitation at L1 from 1200 to 1500 UTC on 4 July 2013. (**a**) The 0.1-mm threshold, (**b**) 1-mm threshold and (**c**) 5-mm threshold. In each performance diagram plot, the lower-left corner represents no forecast skill, and similarly, the upper-right corner indicates perfect skill. The purple curves indicate the critical success index (CSI), and the black dashed lines indicate the frequency bias. The colored dots denote the results of the experiments with the legends shown at the bottom of the figure, and the number inside each dot indicates the forecast time in hours.

## 5. Summary and Conclusions

Influenced by the topography of Jayer Mountain, a moderate precipitation event occurred along the western margin of the Junggar Basin in Xinjiang. In this study, lightning data were assimilated into a numerical model to improve precipitation forecasting. A lightning assimilation scheme based on pseudo-water vapor and driven by the gridded lightning frequency, commonly applied in monsoon regions with frequent strong convention, was tested in the arid inland Xinjiang region. Numerical simulations based on lightning data assimilation were performed to better understand the development mechanism and physical process of leeward slope moderate precipitation.

The results revealed that by assimilating pseudo-water vapor data created based on lightning data, a larger increment in relative humidity was obtained in the analysis field at the location of lightning occurrence, and the largest increment was achieved in the cycling analysis experiment. In the 1-h forecast field, the overland flow from Jayer Mountain in the assimilation experiments was strengthened. Near the lightning occurrence location, the lower layers produced weaker convergence, and the temperature decreased. The assimilation experiments indicated an increase in water vapor on the leeward slope, with positive and negative concomitant potential vorticity anomalies and a distinct upward movement at the downhill position. As the thermal and dynamic fields within the forecast field were varied, the content of ice-phase and rain particles increased in the assimilation experiments. Lightning data assimilation provided suitable water vapor conditions for precipitation occurrence in the model, and the assimilation experiments attained a significantly improved precipitation forecast accuracy in the Jayer

Mountain region, both in terms of subjective analysis and objective forecast skill scores. The improvement effect of the cycling analysis experiment was better than that of the single analysis experiment.

In this study, since the 3DVAR method relies on a static background error covariance matrix, the relative humidity increments exhibit anisotropic characteristics, and thermal and dynamic adjustments of the analysis fields are lacking due to the uncorrelated nature of the control variables. Therefore, in future studies, the application of more advanced assimilation methods, such as a 4DVAR model, ensemble Kalman filter (EnKF) or hybrid method, may further improve the effectiveness of lightning assimilation in heavy precipitation forecasting.

**Author Contributions:** Conceptualization, P.L. and Y.Y.; investigation, P.L.; data curation, P.L.; methodology, P.L. and Y.Y.; project administration, Y.Y.; writing—original draft, P.L.; writing—review and editing, P.L., Y.Y., Y.X. and C.W. All authors have read and agreed to the published version of the manuscript.

**Funding:** This work was supported by the National Natural Science Foundation of China (No. 42175064), the Major Science and Technology Project of Gansu Province (No. 20ZD7FA005) and the National Natural Science Foundation of China (No. 41175092).

**Institutional Review Board Statement:** Not applicable.

**Informed Consent Statement:** Not applicable.

**Acknowledgments:** This work was supported by the Supercomputing Center of Lanzhou University. The authors thank the Chinese National Meteorological Information Center (NMIC) for providing lightning data and the precipitation product (http://data.cma.cn/data/cdcdetail/dataCode/SEVP_CLI_CHN_MERGE_CMP_PRE_HOUR_GRID_0.10.html, accessed on 4 July 2013). The NCEP FNL analysis data are available from the NCAR Research Data Archive (RDA, https://rda.ucar.edu/datasets/ds083.2/#access, accessed on 4 July 2013).

**Conflicts of Interest:** The authors declare no conflict of interest.

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
