# Peer review of "Impact of Lightning Data Assimilation on Forecasts of a Leeward Slope Precipitation Event in the Western Margin of the Junggar Basin"

_remotesensing, doi:10.3390/rs13183584_

Round 1

Reviewer 1 Report

The paper is focused on the impact of lightning data assimilation at the short-term on the forecast of a heavy precipitation event. The forecasts are performed with WRF model and using a 3D-Var method. It is found that assimilating lightning measurement improves the short-term precipitation forecast accuracy compared with the control run.  The manuscript addresses an interesting topic. However the presentation of the study is not sufficiently clear in some points and some important aspects need to be addressed. My recommendation is for major revision. Please find the comments below.

Major comments

-Line 18 (and many other times) “lighting” should be “lightning”. This typo is repeated many times in the paper. All occurrences need to be corrected.

-Lines 34-35 “The annual precipitation below 100 mm occurs in more than 50% of Xinjiang”. Maybe the authors mean that in more than 50% of the Xinjiang region the average annual precipitation is lower than 100 mm. Please rephrase the sentence.

-Line 43 “the hourly accumulative precipitation exceeding 10 mm”. Again, do the author mean “an hourly accumulated precipitation of more than 10 mm”? Please rephrase.

Lines 54-56 It would be helpful to give a general framework of the usefulness of lightning data assimilation at the short-term in the precipitation forecast improvement, briefly describing the main findings of the papers referenced in line 56, and including more studies.

-Lines 126 “values” should be “matrix”

-Lines 132-136 Authors should clarify if they have done sensitivity tests with different choices of the observation error.

-Lines 142-144 “Figure 1 shows the location where the accumulated lighting locations at 1100 UTC and 1200 UTC on 4 July 2013”. The meaning of this sentence is not clear. Please check this sentence and rephrase it.

-Lines 161-168. The experimental set-up is not clearly illustrated. Does the single analysis experiment (LDA) run start at 06:00 UTC as the CTL run?  What about the CLDA? Does the CLDA experiment start at 06:00 UTC too, with a spin-up of 5 hours before the first assimilation cycle? Furthermore, it is not clear how long the experiment lasts. From the first results seems that only one hour of forecast is done after the assimilation phase, since results from 12:00 to 13:00 UTC are presented. In section 4.3, results are then presented for a 3h forecast after assimilation. The experiments should be described better in section 3.1, illustrating the difference between single and cycling analysis. Maybe a table can be added to explain when each experiment starts, how long it lasts and when assimilation is performed for both single and cycling analysis.

-Lines 173-175 The jump between the NCEP-FNL (1°x1°) and the WRF horizontal resolution is too high (with a factor of more than 30 between the horizontal resolution of the NCEP final analysis and the WRF horizontal grid resolution). I recommend the authors to do an additional numerical experiment with a parent grid of the grid used in this paper to confirm the results. A nesting factor of 3 or 4 between the WRF grids is recommended.

-Figure 4: Is the color bar of the upper figures (a, b and c) the same of the lower ones (d, e and f)?

-Lines 278-279: “the most relevant to precipitation” should be “the most relevant for precipitation”

-Section 4.2 and 4.3 have the same title. Please check.

-In the Introduction, authors describe the case study as a heavy precipitation event with hourly precipitation higher than 10 mm in a region where the annual average precipitation is about 110 mm. Results are then presented for the first hour of forecast after the assimilation phase (i.e. 12:00 – 13:00 UTC) and for a 3h-forecast (i.e. 12:00-15:00 UTC). It is not clear if the hourly precipitation the authors mention (10 mm) is referred to the first hour of forecast, or a similar amount is recorded also for the following hour. It would be useful to explain better this point, reporting the precipitation for each of the three hours.

-Line 329 what do the authors mean by “two hours before the forecast”. Isn’t the period 12:00-14:00 already forecast time? This point need to be clarified better.

-In Figure 10 the CLDA experiment is reported in red while the LDA experiment is in blue, but in Figure 11 the colors are reversed. This can be confusing. I suggest to maintain for Figure 11 the same color assignments of Figure 10.

Minor comments

Line 92 “assimilated” should be “when assimilating”

Line 141 “nearest model grid” should be “nearest model grid point”

Line 141 “the” should be “then”

-Titles of Figures 2a and 2b are cutted in my copy of the paper.

Reviewer 2 Report

In attachment my comments. 

Round 2

Reviewer 1 Report

The manuscript has been improved and can be accepted for publication in its current form.

Author Response

Thank you for your work of manuscript.